# OpenReview forum: "Sampling from Your Language Model One Byte at a Time"
_ICML.cc/2026/Conference — ICML 2026 regular_

### Official Review · Reviewer_xZvy · 2026-02-16

**Soundness:** 2
**Presentation:** 2
**Significance:** 2
**Originality:** 2
**Overall Recommendation:** 4
**Confidence:** 4

**Summary:**

​​In this paper, the authors propose a solution to the Token Boundary Problem in LLMs, which arises when the prompt and the subsequent completion do not form a valid token sequence. The authors cite cases where this can occur as a result of the prompt being tokenized with tokens that are not encountered by the model in the tokenized corpus during training. To address this, the authors use the concept of Valid Covering Trees, an extension of the Covering Trees introduced by Vieira et al. 2024, restricted to valid token sequences. The authors propose to use this construction to sample valid completion, and along the way, use this to construct next-byte distributions for an LLM. The paper compares their method with other approaches to solving the PBP in terms of overhead, evaluates the next-byte prediction accuracy of the method, and applies it to the problem of ensembling models with different tokenizers and to proxy-tuning.

**Compliance With Llm Reviewing Policy:**

Affirmed.

**Final Justification:**

Given the author's rebuttal, I increased my score because the experimental results are interesting and valuable. However, I still have quite a few concerns:
- The paper does not convincingly demonstrate that the BS offers any improvement in real settings.
- The technical contributions are very limited; Proposition 3.1 is not novel, and this is not clearly discussed in the paper.
- The paper, in my opinion, needs significant improvements when it comes to exposition, writing, and rigour.

Currently, I find it hard to assess if such concerns outweigh the value of the experiments.

**Key Questions For Authors:**

Below are my main questions for the authors. I will increase my score if the answers are satisfactory.

- It is unclear to me the role that Proposition 3.1 plays in the paper. I might have missed how/where it is used, but I am not able to find a pointer that references Proposition 3.1, and the paragraphs before and after section 3.1 do not really clarify this either. Is the Proposition needed to construct the VCT? If so, it should be explicitly stated.

- I am not able to find any clear comparison in benchmarks between the accuracy of the model using ByteSampler versus standard inference. For example, why are the tasks in Table 7 not used to compare the base models with ByteSampler inference? This makes it hard to decide if ByteSampler degrades the performance of the model or not. I am aware that Table 6 compares next-character predictions between ByteSampler and plain BPE, but how does this translate to accuracy in a benchmark?

- Similarly, in Section 4.1, you compare the sampling speed versus other PBP methods, but never state how slower ByteSampler is compared with standard decoding. Overall, this makes it hard to decide if ByteSampling is viable to use or not.

- Are the ensemble results in Table 7 comparing the accuracy of the byte-level ensemble with the plain BPE accuracy of each model, or with the accuracy of running the model with the ByteSampler algorithm? This is not clearly stated.

Please let me know if I have missed anything in the paper that answers my concerns.

**Limitations:**

- The authors have cited multiple references regarding the causes of the PBP. However, it is unclear to what extent this problem has a real and significant impact on the LLM's output quality. Consequently, for a reader, I believe it is hard to gauge if the method proposed by the authors, which requires changing inference pipelines, is "worth" in terms of improving the user experience. If there are references discussing the real impact of PBP, they should be cited.

- As mentioned in the "Key questions" section, in my opinion, the lack of comparisons between ByteSampler and standard decoding (in terms of decoding speed and accuracy) is a limitation of the work. Please, if I have missed such comparisons, let me know.

- The technical novelty of the paper seems limited. Given the similarity with Vieira et al. 2024 regarding the covering tree, the author's method seems like an extension of theirsmethod. Even though Vieira et al. 2024 is properly discussed, it could be helpful if the authors could comment more clearly on their contributions.

**Strengths And Weaknesses:**

I will detail both strengths and weaknesses separately.

**Strengths:**
- I believe there are very positive aspects in the way the authors present their work. This includes, for instance, the use of colors to help the reader identify which objects are in token space and which are in string space. The examples given throughout the paper make the presentation engaging and help ground and contextualize the problem of PBP (eg, the list in line 80, the table 1, Figure 2 and 3, etc.)

- I believe the formalism is sound and correctly extends the Covering Trees introduced by Vieira et al. 2024. The solution proposed to the PBP and the construction of next-byte distributions is insightful and relevant.

- The authors also present ample experiments and empirical evidence on the performance and utility of ByteSampler. The authors convincingly show that their method has less overhead than other alternatives by Vieira et al. and by Phan et al..

- The application to model ensembling when the LLMs have different tokenizers is relevant, since previous attempts were mainly based on empirical constructions (eg, using optimal transport or mapping between vocabularies, eg  Xu et al 2024, *Bridging the Gap between Different Vocabularies for LLM Ensemble*).



**Weaknesses:**
- There are some aspects of the presentation that need to be updated and improved. I cite a few suggestions. In footnote 1, it could be useful to clarify that the functions encode and decode are defined by the tokenizer. The term "(in)valid token sequence" is used before being defined (e.g., you use it in line 147, and then include its definition later in Proposition 3.1). Additionally, is there a reason to use the term "valid" instead of "canonical"? Multiple papers use the second one, such as Vierira et al 2024, Geh et al 2025, or A. Velasco et al 2025.

- It is unclear to me the role that Proposition 3.1 plays in the paper. See the "Key questions" section.

- There are some empirical evaluations and details that, in my opinion, are missing. See the "Limitations" section.

- The space corresponding to the BPE inference in Alg 1 could be used for more relevant information, since BPE is rather standard.

- Given that the main technical object of the paper is based on the VCT, I find it rather strange that the authors do not give a precise statement on how to construct it in full generality. Instead, Fig 2 only gives a specific example.

- Some terms are not clearly defined in the experiment section. For instance, in line 245, what does "how many inference tokens ... each method needs to solve BPB" mean?

- There are some minor typos throughout the paper. Some are grammatical, such as "the" in line 1162, which I believe should be "that"; in line 242 I believe you discussed the overhead in Section 4.1, not 4.2, and in footnote 3, line 108, there is an "is" missing. However, others are "structural". For instance, in the proof of Proposition 3.1, you use the notation S_1, S_2, and T_1 and T_2, but these are not defined within the proof (I believe they refer to the notation from Lemma D.2, but this is unclear and confusing). Furthermore, in such proof, you claim things that are not apparently correct, or at least are not trivial. In particular, why is  "This merge can also be applied when encoding S_1 + S_2" a correct statement? When encoding the full string decode(t_1,...t_n), isn't it possible that some previous merge renders the merge crossing the boundary of t_i and t_{i+1} impossible, eg, if t_i is merged with t_{i-1} previously?

---

> ### Author Rebuttal · Authors · 2026-03-31
>
> ## Invalid token sequences
> We will clarify the definition of encode and decode, and ensure that the concept of token sequence validity is defined before it is used. Our use of the valid/invalid terminology for token sequences follows Phan et al. (2024) and (2025) which use the same definition.
>
> ## Proposition 3.1
>
> Proposition 3.1 is required to argue that the edge-wise filtering we do in Figure 2c is correct in the sense that it filters out exactly the invalid paths in the tree. The routine described in Section 3.1 is used in line 262 (left) in Algorithm 2 to filter the tree for valid next tokens. We agree that this is not adequately explained and will make sure to clarify this in the revision.
>
> We agree that the proof of Proposition 3.1 was incomplete as written. If we assume (WLOG) that the $[t_i, t_{i+1}]$ is the invalid token pair with the earliest boundary crossing merge M’ (during the tokenization of decode$([t_i, t_{i+1}])$), then the statement is valid. We will include the amended proof in the revision.
>
> ## VCT Construction
>
> (Please see our response to reviewer REEg _"Clarity in Section 3"_)
>
> ## Clarifying experimental terms
>
> By “how many inference tokens ... each method needs to solve BPB” we are referring to the number of inference tokens that method must evaluate the language model on. This directly measures the method’s cost in terms of model forward passes. We will make this clear in the revision.
>
> ## Accuracy on benchmarks
>
> Regular evals are formatted specifically to not have the PBP. For instance, a prompt like “Answer:” basically guarantees that the prompt does not end with a partial token w.r.t. to any reasonable continuation, since the next character is effectively guaranteed to be whitespace, and all major models pretokenize on whitespace. In these cases, applying ByteSampler will have no effect since the PBP is not an issue to begin with.
>
> When the PBP is an issue, we show that ByteSampler improves performance in Appendix E.4, where we report results on HumanEval Random Span, a code completion benchmark where the prompt prefixes are uniformly selected. In this setting, ByteSampler outperforms normal sampling by a significant margin (+26 pp).
>
>
> ## Wall-clock overhead compared to normal sampling
>
> (Please see our response to reviewer vEd4. _"Wall-clock overhead compared to normal sampling"_)
>
> ## Ensemble results
>
> In the ensemble results, the individual models are also run using ByteSampler to isolate the effect of the ensembling. However, we expect the performance to be identical because the benchmarks have been carefully formatted to avoid prompt boundary issues.
>
> ## Practical Impact of the Prompt Boundary Problem
>
> While the prompt boundary problem may be easy to avoid in English, where words are small and delimited by whitespace, there are many settings where the PBP can arise naturally and unbeknownst to the user. Recently, Xu et al. 2026 [1] studies how the PBP can naturally arise in domains where whitespace does not reliably separate semantic units: languages that do not use whitespace (e.g., Chinese), highly compounding languages (e.g., German), and non-natural languages (e.g., code). In these domains, user prompts that end with a complete word or syntactic unit may nonetheless end with a partial token relative to the desired completion, causing a serious failure mode in models from various families and sizes (including, e.g., Qwen-32B). In analysis, they further find that the PBP does not improve with model scale, pointing to the continued value of methods like ByteSampler.
>
> [1] Xu, Hao, et al. "Are you going to finish that? A Practical Study of the Partial Token Problem." arXiv e-prints (2026): arXiv-2601.
>
> ## Technical Novelty
>
> While our method can indeed be viewed as an extension of Vieira et al. 2024, we make several significant contributions which believe justify the novelty of our paper:
>
> 1. Pruning of the tree using the structure of BPE (Section 3.1). Although these techniques are not novel by themselves, they have not been applied to the PBP or to byte/char-level sampling previously.
> 2. We explicitly model the pretokenizer (See Appendix C.3) allowing us to account for its instability with respect to future bytes. This is crucial to make the pruning system work for many popular models and is a key advantage over existing methods like Phan et al., 2025 and Turaga, 2025. The issues posed by pretokenization regexes are quite subtle and we are not aware of any previous treatment of them in the literature. This makes ByteSampler the first method which is both exact, efficient, and supports the vast majority of models in use today. We will emphasize this more in the revision.
> 3. We note that applying sampling parameters like temperature at the byte-level has a different effect than applying them at the token level and extend our method to support sampling transforms at token level even during byte-level sampling in Appendix F. We are the first to point out this issue and address it.

---

> > ### Author Rebuttal · Reviewer_xZvy · 2026-03-31
> >
> > Thank you for your response.
> >
> > I still have quite a few reservations:
> >
> > - Importance of PBP. Thank you for the reference. I understand that in the contexts you mention (i.e., non-English languages, code, etc.), the PBP can be more pronounced. However, I do not think your response addresses my concern, which is that in the current version of the paper, it is hard to understand the benefit a user would see when running a standard LLM for a task like coding versus correcting for the PBP. Currently (please let me know if I missed it), the paper does not present results showing the practical benefits of ByteSampler in a task like coding, where the authors state that “In these domains,…, causing a serious failure mode.”
> >
> > - Ensemble results. The authors claim that: "In the ensemble results, the individual models are also run using ByteSampler to isolate the effect of the ensembling. However, we expect the performance to be identical because the benchmarks have been carefully formatted to avoid prompt boundary issues." Given this, I do not fully understand why the authors do not present the accuracy of the base models with standard sampling. If you expect the performance to be identical, then it would be useful to confirm it and present it in the paper.
> >
> > - In general, I still have reservations regarding the novelty of the work, even after reading the authors’ rebuttal. This concern is further exacerbated by the fact that: (i) as far as I understand, the main technical contribution is based on Proposition 3.1, and, as the authors confirmed, its proof was not entirely sound (to be clear, I agree that Proposition 3.1 holds, but I believe it is concerning that it was not carefully checked in the original manuscript) and (ii) the construction presented in Section 3 around the VCT is very informal.

---

> > > ### Author Response · Authors · 2026-04-03
> > >
> > > We thank the reviewer for their continued thoughtful engagement with our work!
> > >
> > > ## Impact of prompt boundary problem
> > >
> > > To illustrate the impact of the PBP in realistic code settings, in Appendix E.4 we report results on HumanEval Random Span, where the prompt is a uniformly selected prefix of the gold solution. This is more likely to expose the PBP, as the prompt ending could be anywhere, including in the middle of variable and function names. In this setting, ByteSampler outperforms normal sampling by a significant margin (+26 pp). We reproduce those results here for convenience.
> > >
> > > | Method | pass@1 |
> > > | :- | -: |
> > > | Naive | 0.565 |
> > > | 1 Token Backtracking (Token Healing) | 0.716 |
> > > | 2 Token Backtracking (Token Healing)  | 0.741 |
> > > | 4 Token Backtracking (Token Healing)  | 0.738 |
> > > | ByteSampler (ours) | 0.824 |
> > >
> > > ## Standard sampling results
> > >
> > > Unfortunately we were not able to run these experiments in time for the main response, but they are ready now. We appreciate your patience while we rewrote our eval harness to support both token and byte-level sampling. The results obtained show that ByteSampler (in byte-level sampling mode) achieves similar average performance to the HuggingFace ("standard sampling") baseline over the full set of evals:
> > >
> > > | Sampling method | HF greedy | BS greedy | HF temp1 | BS temp1 |
> > > | --- | ---: | ---: | ---: | ---: |
> > > | Average performance (all models all datasets) | 0.563  |  0.584  |  0.464 | 0.456 |
> > >
> > > These results suggest that the average performance of BS "temp1" (standard sampling at temperature 1) is very similar to that of HF temp1, although there is some positive and negative variation among tasks. On the other hand, switching to greedy (which has no expectation of equivalence) shows an advantage for ByteSampler.
> > >
> > > As far as we know, the prompt boundary problem should not significantly influence these results due to the formatting used.
> > >
> > > We give the individual results below.
> > >
> > > ### Llama 3.2 1B
> > >
> > > | Benchmark | HF greedy | HF temp1 | BS greedy | BS temp1 | n |
> > > | --- | ---: | ---: | ---: | ---: | ---: |
> > > | squad | 0.6918 | 0.5854 | 0.6936 | 0.5116 | 5000 |
> > > | lambada | 0.5218 | 0.3002 | 0.5148 | 0.2296 | 5000 |
> > > | triviaqa | 0.4478 | 0.3344 | 0.4430 | 0.2616 | 5000 |
> > > | wikidataqa | 0.6646 | 0.4608 | 0.6568 | 0.4674 | 5000 |
> > > | jeopardy | 0.2738 | 0.1894 | 0.2642 | 0.1422 | 5000 |
> > > | arithmetic | 0.8357 | 0.7923 | 0.8337 | 0.7617 | 4998 |
> > > | drop | 0.2954 | 0.2214 | 0.2986 | 0.1876 | 5000 |
> > > | avg | 0.5330 | 0.4120 | 0.5292 | 0.3660 | — |
> > >
> > > ### Olmo2 1B
> > >
> > > | Benchmark | HF greedy | HF temp1 | BS greedy | BS temp1 | n |
> > > | --- | ---: | ---: | ---: | ---: | ---: |
> > > | squad | 0.7716 | 0.6544 | 0.8024 | 0.6458 | 5000 |
> > > | lambada | 0.5356 | 0.3946 | 0.7156 | 0.4468 | 5000 |
> > > | triviaqa | 0.5178 | 0.3818 | 0.5346 | 0.3528 | 5000 |
> > > | wikidataqa | 0.6908 | 0.2114 | 0.6886 | 0.5302 | 5000 |
> > > | jeopardy | 0.3116 | 0.2118 | 0.3266 | 0.1870 | 5000 |
> > > | arithmetic | 0.8475 | 0.8095 | 0.8445 | 0.8003 | 4998 |
> > > | drop | 0.3648 | 0.2886 | 0.4094 | 0.2734 | 5000 |
> > > | avg | 0.5771 | 0.4217 | 0.6174 | 0.4623 | — |
> > >
> > > ### Qwen3 1.7B
> > >
> > > | Benchmark | HF greedy | HF temp1 | BS greedy | BS temp1 | n |
> > > | --- | ---: | ---: | ---: | ---: | ---: |
> > > | squad | 0.8084 | 0.7982 | 0.8446 | 0.8026 | 5000 |
> > > | lambada | 0.6132 | 0.5828 | 0.6116 | 0.5416 | 5000 |
> > > | triviaqa | 0.3356 | 0.3056 | 0.3888 | 0.2866 | 5000 |
> > > | wikidataqa | 0.6596 | 0.6302 | 0.6578 | 0.6138 | 5000 |
> > > | jeopardy | 0.2128 | 0.1854 | 0.2744 | 0.1738 | 5000 |
> > > | arithmetic | 0.9796 | 0.9782 | 0.9798 | 0.9722 | 4998 |
> > > | drop | 0.4432 | 0.4262 | 0.4700 | 0.3942 | 5000 |
> > > | avg | 0.5789 | 0.5581 | 0.6039 | 0.5407 | — |
> > >
> > > ## Proposition 3.1 and Contributions
> > >
> > > We include a proof of Proposition 3.1 for completeness, but the result has been proven previously by e.g. van Antwerpen & Neubeck (2025). We will amend the proof accordingly while also improving the presentation of the VCT. See also our rebuttal reply comment to reviewer REEg, where we write out the proof of the Proposition D.1 (which is specific to the VCT we construct) in a more explicit way.
> > >
> > > We consider our main contributions to be the application of this result to solve the PBP as well as the correct and general handling of layers other than BPE (e.g. pretokenization, special tokens, normalization). Additionally we provide an implementation that, while not as fast as normal sampling, is dramatically faster than all competitors at the same quality. This makes byte-level sampling more practical because it can be applied to models that are popular today and with reasonable performance and negligible memory overhead.

---

### Official Review · Reviewer_vEd4 · 2026-03-10

**Soundness:** 4
**Presentation:** 4
**Significance:** 3
**Originality:** 3
**Overall Recommendation:** 5
**Confidence:** 4

**Summary:**

This paper proposes ByteSampler, an inference-time algorithm that allows autoregressive language models trained with BPE tokenization to generate text one byte (or character) at a time while preserving the original model distribution. The motivation is the Prompt Boundary Problem (PBP), where tokenization prevents tokens from crossing the boundary between prompt and completion, which can distort generation probabilities. The method constructs a Valid Covering Tree that represents all valid token sequences compatible with a given byte prefix and uses it to compute exact byte-level probabilities while maintaining constant-time complexity with respect to prefix length.

The authors show that this approach eliminates the prompt boundary problem, preserves language modeling quality, and enables additional applications such as byte-level ensembling across models with different tokenizers and proxy-tuning across models with mismatched vocabularies.

**Compliance With Llm Reviewing Policy:**

Affirmed.

**Final Justification:**

My opinion on this work does not change after rebuttals, not game changing enough to warrant a 6 but very well executed and rigorous. It fully has its place at ICML in my opinion.

**Key Questions For Authors:**

- As mentioned in the conclusion, one potential application of this method is cross-tokenizer distillation. Did you explore this direction in practical settings to validate that the approach works in practice? This would likely be beyond the scope of the current paper, but it seems like a natural and compelling application.

- Could the authors provide more details on the overhead introduced by the method in terms of wall-clock speed and memory compared to the baseline model?

**Limitations:**

yes

**Strengths And Weaknesses:**

# Strengths

- The paper addresses a well-known but often underappreciated issue in tokenized language models: the prompt boundary problem. The problem is clearly explained and illustrated with concrete examples.

- The proposed method is conceptually elegant. The Valid Covering Tree formulation provides a principled way to enumerate valid token sequences compatible with a byte prefix while avoiding the exponential explosion seen in prior exact methods.

- Experimental results support the claims that the method preserves language modeling quality and avoids the degradation caused by naive character-level prediction.

- The paper demonstrates interesting applications beyond simply fixing the prompt boundary issue, including ensembling models with mismatched tokenizers and transferring post-training recipes across models through proxy-tuning.

- The work is extremely well written, technically thorough, and a pleasure to read.


# Weaknesses

- The paper positions ByteSampler primarily as a solution to the prompt boundary problem. In practice, this issue is often mitigated through preprocessing, prompt formatting, or chat templates, so the main framing may not feel entirely game-changing. The cross-tokenizer applications, on the other hand, seem to be the most interesting aspect of the work.

- Many of the applications presented (such as model ensembling and proxy-tuning across tokenizers) are promising but somewhat exploratory. The experiments demonstrate feasibility but do not fully characterize the trade-offs or limitations of these use cases. Cross-tokenizer distillation is mentioned but not explored, and this is arguably the most compelling potential application. Seeing this investigated would have made the contribution (even) stronger.

- Most importantly, I did not see clear discussion of the overhead in terms of wall-clock time or memory introduced by this method. The paper reports relative overhead compared to other mitigation techniques, but overhead relative to the vanilla baseline model is crucial. In addition, it would be useful to understand whether this approach interferes with efficient inference frameworks used in practice (e.g., vLLM-style optimizations). What would the overhead look like in a realistic inference setup? Similarly, what would the implications be in a training setting if the method were used for tasks such as cross-tokenizer distillation?

- The discussion of pretokenization currently appears mostly in the appendix, but it is a key component of modern tokenization pipelines and something I was actively looking for while reading the paper. Moving part of this discussion to the main text would improve clarity. Relatedly, it would help to be explicit about the practical trade-offs involved, including any computational overhead introduced by handling these pre-tokenizer features, which would slightly nuance the current "free win" impression.

---

> ### Author Rebuttal · Authors · 2026-03-31
>
> We would like to thank the reviewer for the careful review and address each of the concerns raised below. We are also eager to address any remaining concerns during the discussion period!
>
> ## ByteSampler for distillation
>
> We agree that cross-tokenizer distillation is an exciting application of ByteSampler, and in fact we are actively working on this direction. As the reviewer suggests, it is beyond the scope of the current paper, as additional work is required to make byte-level distillation practical. However, we hope to publish a follow-up with these results soon.
>
> As an example, byte-level distillation requires optimized “prefill oriented” routines which can evaluate the byte-level distributions at every byte in a sequence efficiently. Among other things, this requires attention kernels that can handle large unstructured trees, which are not needed during generation (as the trees used for generation are generally tiny). Additionally, the current ByteSampler “post processing” algorithm in Paragraph 3.3.3 does not vectorize naturally over the sequence direction, so we develop a method to compute the same thing efficiently in a vectorizable manner.
>
> ## Wall-clock overhead compared to normal sampling
>
> This is an excellent suggestion and we have run experiments to address this which we will include in the revision. To benchmark the speed of our method, we sampled 512 byte completions of a 127 token / 636 byte prompt using Llama-3.1-8B, collecting 30 samples each.
>
> | Method | Total Time (s) | Sampling speed | Ratio |
> | - | - | - | - |
> | Normal sampling | 2.85 | 182.7 ± 1.8 | 1.00x |
> | ByteSampler PBP Mode | 2.84 | 182.5 ± 1.8 | 1.00x |
> | ByteSampler Byte Sampling | 13.7 | 37.1 ± 0.2 | 0.21x |
>
> These results reflect the fact that simply fixing the PBP can be done with negligible overhead, while byte-level sampling is indeed significantly slower than normal sampling. This is because we must invoke the LM at least once for every byte (instead of once per token) so we will naturally be at least ~4.5x slower than normal sampling since the average number of bytes per token is roughly 4.5. In that light, we are very close to the theoretical best performance possible for an exact method (the remaining gap is due to the overhead in the tree construction).
>
> | Method | Total Time (s) | Sampling speed |
> | - | - | - |
> | ByteSampler Byte Sampling | 13.7 | 37.1 ± 0.2 |
> | ByteSampler VLLM | 14.5 | 35.1 ± 0.3 |
>
> While VLLM does run the model in an optimized manner for inference, it is poorly suited to inference on trees, due to its blocked prompt caching. We are working on tree-structured flash attention kernels which could be used to bring native tree inference to VLLM, but this is beyond the scope of this paper for now.
>
> ## Pretokenization
>
> We will make the discussion of pretokenization more prominent in the revision, as we do consider it a major contribution of our work. The overhead paid due to pretokenization depends on the pretokenizer, but all of the tokenizers we are aware of pay (at worst) the following costs:
>
>
> 1. Either 2 extra token evaluations near apostrophes to handle contractions.
> 2. 1 extra token for runs of multiple whitespace, to handle the whitespace lookahead.
>
> So the additional costs are not overly harsh. We will add this information along with aggregate statistics to the appendix.

---

> > ### Author Rebuttal · Reviewer_vEd4 · 2026-03-31
> >
> > I appreciate the new results and agree my other points are out of scope. Eager to read the follow up work.
> >
> > My opinion on this work does not change, not game changing enough to warrant a 6 but very well executed and rigorous.

---

### Official Review · Reviewer_VdNQ · 2026-03-13

**Soundness:** 4
**Presentation:** 4
**Significance:** 4
**Originality:** 4
**Overall Recommendation:** 6
**Confidence:** 1

**Summary:**

The author present a method to convert any LM with a BPE tokenizer into a byte-level language model. The method works at inference time, solves the prompt boundary problem and allows ensembling models with different tokenizers.

**Compliance With Llm Reviewing Policy:**

Affirmed.

**Final Justification:**

I thank the authors for their comprehensive rebuttal. I believe this is an interesting and impactful paper and I will keep my positive score.

**Key Questions For Authors:**

N/A

**Limitations:**

yes

**Strengths And Weaknesses:**

# Strengths

The proposed approach uses the BPE algorithm's properties to design a byte level sampler for language models. The approach seems theoretically sound and the evaluation methodology used is appropriately motivated. Specifically reporting of Bits Per character and inference overhead are sound ways to evaluate the approach.

Crucially it solves the prompt boundary problem where the prompt may be cutoff at arbitrary character unrelated to the token boundary. The results show that the proposed approach recovers the performance of the model under such kind of prompt boundary noise. The overhead is reasonable given that the approach completely recovers the baseline performance. Thus this is a significant contribution.

The presentation is well structured and the narrative is easy to follow.

In terms of originality, it is a highly novel contribution with a novel approach to byte level sampling from pretrained language models. The application of the approach to ensembling and post-training is also highly novel and significant.

# Weakness
 The presentation in section 3 could be improved by starting with section 3.3 i.e. starting with what is required to achieve the goals in section 3.3 and then building up the methodology to show how it achieves those goal. This would be especially helpful for readers who are unfamiliar with the details of the nuances of the BPE algorithm.

---

> ### Author Rebuttal · Authors · 2026-03-31
>
> We would like to thank the reviewer for their positive review! We have addressed the reviewer's concern below and are happy to answer any other questions about our work.
>
> ## Section 3 presentation
>
> Thank you for your suggestions on how to reorganize section 3 to clarify the overall logical flow. We will revise the presentation accordingly. In particular, we plan to make the current “VCT intro” into a proper subsection and add a new intro which outlines the overall goals and how the subsections fit together to achieve those goals, as well as referencing relevant sections in the appendix for more details.

---

> > ### Author Rebuttal · Reviewer_VdNQ · 2026-03-31
> >
> > My concerns have been adequately addressed.

---

### Official Review · Reviewer_REEg · 2026-03-16

**Soundness:** 2
**Presentation:** 3
**Significance:** 3
**Originality:** 2
**Overall Recommendation:** 4
**Confidence:** 3

**Summary:**

This paper proposes an inference-time approach to sampling a sequence of bytes from an arbitrary language model. The key challenge is to ensure the "exactness" of the sampler, i.e. generating only valid/canonical sequences, and in accordance with the distribution over bytes induced by the LLM's distribution over tokens, while maintaining computational efficiency. The authors propose a method based on constructing and incrementally updating a valid covering tree (VCT) that captures the possible token sequences given the current byte prefix, and uses this to compute the byte distribution. The paper shows how the byte-level sampler can be used to fix the prompt boundary problem, as well as be used to combine and adjust langauge models with different tokenizers.

**Compliance With Llm Reviewing Policy:**

Affirmed.

**Final Justification:**

I appreciate the authors' response, which addressed some concerns. The paper remains borderline for me due to the need for significant changes in presentation to make clear the novel methodological contribution in a revision. I have increased my score to be on the positive side as I tend to agree with the authors' argument that the application to byte-level sampling is novel and interesting, even if it could be further developed.

**Key Questions For Authors:**

- Why do the authors consider only smaller language models (around 1B params)? In particular, do the authors envisage use cases for their method on larger models (where perhaps the PBP would be less problematic)?

Please also address the weaknesses above.

**Limitations:**

yes

**Strengths And Weaknesses:**

Strengths:
- The paper introduces a novel approach to constructing byte-level models that is much more efficient that existing methods with guarantees.
- There is a clear demonstration of the utility of the byte-level models through merging and combining LMs with different tokenizers.
- The experiments convincingly demonstrate that the byte-level models solve the prompt boundary problem for smaller models such as OLMo2-1B and Qwen3-1.7B.

Weaknesses:
- Prop 3.1 seems very similar in nature to Theorem 1 in (Vieira et al. 2025) which also reduces canonicity/validity to bigrams.
- There is not sufficient clarity or rigor in the presentation of the valid covering tree idea. There is no formal definition of VCT nor a statement or proof of the "compactness" property involving a trunk + branching nodes.
- The claim that "we do not expect there to be any significant differences between them in practice" (Start of Section 4.1) is not justified. Given that they are not theoretically equivalent there should be some quantitative evaluation of quality.

---

> ### Author Rebuttal · Authors · 2026-03-31
>
> We thank the reviewer for the thoughtful review. We address each of their concerns and questions below and are eager to discuss further.
>
> ## Novelty of Prop 3.1 relative to past work
> We apologize for any confusion that we may have caused. As we note in the footnote on line 163 (right), the pairwise proposition was also given previously in van Antwerpen & Neubeck (2025). The main goal of this subsection is to familiarize readers with this fact, since it is not very well known. We will make this relationship more clear and also mention the Vieira et al. 2025 paper.
>
> ## Clarity in Section 3
> We give the steps to construct the VCT in the caption of Figure 2. We believe these steps unambiguously specify the tree, although it reads informally. We also give an algorithm to incrementally construct the tree in Algorithm 2, although this may not make its structure clear. To address this, we will add a precise description in the main text to describe the tree’s structure explicitly.
> As for the compactness proof, it is given in Appendix D.1, which is referenced in the main text.
>
> ## Differences in quality between PBP methods
> This is an excellent point and we will make the justification for this claim more clear in the rebuttal. The main experiment measuring the performance difference between various PBP methods is the result in Appendix E.2 Table 10, where we evaluate the language modeling loss of the beam summing algorithm of Vieira et al. 2025 in comparison with ByteSampler. What we observe is that we are unable to distinguish them given samples of 1 million byte predictions.
> For the methods of Phan et al. (2025) and Turaga, 2025, it is not currently possible to make this comparison because the set of models supported are disjoint from ours. Both prior works support SentencePiece models like Llama 1/2, the original Mistral models, and Yi. We support a large selection of modern models, including Llama 3/4, Qwen 1/2/3, OLMo 1/2, DeepsSeek V3, and will likely support new models automatically as they are released.
>
> ## Differences in model size
> Recent work [1] has found that the sensitivity of models to the prompt boundary problem does not improve with model scale. Intuitively, this may be because, if anything, larger models are more fit to their tokenizers: given “␣wor”, a larger model is more confident that “ld” is not the correct next-token as the token sequence [␣wor, ld] was never seen in training. As a result, we expect our method for overcoming the PBP to continue to be relevant even in the large model setting.
> Additionally, (and for the same reason) the probability mass assigned to invalid token sequences decreases with model scale, which means that differences between the various exact methods will also decrease at larger scale. Thus, the small model scale we are testing is of interest because it is most sensitive to these differences. To demonstrate this, we reran the experiment of Appendix E.3 for the larger OLMo2 7B model to isolate this effect:
>
> | Model | Token level | Local Canonicalization | Difference  (× 1e-4) |
> | - | - | - | - |
> | OLMo2 1B | 1.056142 | 1.055851 | 2.91 ± 0.18 |
> | OLMo2 7B | 0.792140 | 0.792140 | 2.08 ± 0.11 |
>
> The results show that OLMo2 7B allocates less probability to invalid next tokens, as expected.
> The last reason to run the experiments with small models is due to resource constraints, as large models require larger GPUs, especially when ensembling several of them at once.
>
> [1] Xu, Hao, et al. "Are you going to finish that? A Practical Study of the Partial Token Problem." arXiv e-prints (2026): arXiv-2601.

---

> > ### Author Rebuttal · Reviewer_REEg · 2026-04-03
> >
> > I thank the authors for their response.
> >
> > I am satisfied with the response regarding model size and the response on comparing different PBP methods.
> >
> > I still believe that the fact that there is nowhere a formal, general definition of a VCT is a significant barrier to understanding the paper as-is. As such, I would appreciate a formal writeup of the VCT definition, either through OpenReview or an anonymized link. This would give me more confidence that this issue could be addressed in a revised version.
> >
> > Thank you for pointing to D.1. My concern is that it refers to VCT definition that is not properly defined. This makes it difficult to ascertain how the (very short) proof corresponds to the VCT.

---

> > > ### Author Response · Authors · 2026-04-03
> > >
> > > Thank you for engaging with us! Here is the definition we plan to use (using some Python slicing notation since it is easier to type here).
> > >
> > > ### Definition of the VCT
> > >
> > > The VCT of a string prompt `P` is the tree of all finite token sequences `S` such that:
> > >
> > > 1. `P` is an (inclusive) prefix of `decode(S)`
> > > 2. Last token does not end after the prompt, i.e. `decode(S[:-1])` is an (inclusive) prefix of `P`.
> > > 3. Every contiguous pair of tokens in `S` is valid (according to Proposition 3.1)
> > >
> > > where `decode` is the BPE decoding function of the tokenizer. In our convention, every token sequence begins with the `BOS` token, but the `BOS` token decodes to the empty string, for the purpose of this definition.
> > >
> > > ### Proof of compactness
> > >
> > > Here is a more concrete proof of the compactness, which may be more intuitive than the proof of D.1 at the expense of being a bit longer.
> > >
> > > From _Berglund & van der Merwe (2023)_, we know the "first" branch (i.e. the branch closest to the root) in the `VCT(P)` must start at a position `L ≤ L₀` bytes from the end of `P` (where `L₀` is a constant depending on the tokenizer). We call these first `length(P) - L` bytes the "trunk" and note that they must be the canonical encoding of `P[:-L]` by Proposition 3.1 (since every sequence `S` in `VCT(P)` is valid and every subsequence of a valid sequence is valid as well). Thus, `VCT(P)` can be decomposed into a "trunk" `encode(P[:-L])` attached to "branches" `VCT(P[-L:])`.
> > >
> > > Now, focusing on `VCT(P[-L:])`. Every sequence `S` can be broken into a prefix `S_prefix` plus a last token `s_end`. Consider two sequences `S`, `S'` with `decode(S_prefix) = decode(S'_prefix)`, both of which are prefixes of `P[-L:]`. At most one can be in the `VCT(P[-L:])`, because there is only one valid encoding of `decode(S_prefix)` (and from Proposition 3.1, we know that all sequences in `VCT(P[-L:])` are valid encodings of their contents). Therefore, we can have at most one prefix sequence per byte of `P[-L:]`, which has length `L`. Additionally, the number of final `s_end` tokens per prefix is bounded by the vocabulary size, thus the total number of tokens in `VCT(P[-L:])` is constant.
> > >
> > > ### Note on inference cost for PBP
> > >
> > > It's worth noting that given an `S_prefix` we can score all possible `s_end` continuations in a single LM forward pass (because an LM gives you the probabilities of all next tokens). Therefore, the total number of forward passes needed to evaluate the entire `VCT(P)` tree is `length(encode(P[:-L])) + L ≤ length(encode(P)) + L₀`, so our overhead in token evaluations compared to the standard sampling approach costing `length(encode(P))` is bounded by `L₀`, a constant!

---

### Decision · Program_Chairs · 2026-04-30

**Decision:**

Accept (regular)

**Comment:**

The reviewers recognized the soundness of the proposed inference-time method for byte-level sampling from autoregressive LLMs with BPE tokenizers. Reviewers agreed that the approach addresses the prompt boundary problem (PBP) with low overhead and enables practical applications such as cross-tokenizer model ensembling and proxy-tuning. The construction of a Valid Covering Tree (VCT) for exactness and canonical sequence generation is considered a solid contribution.

The reviewers raised concerns regarding the lack of a compactness proof for the VCT in the main text(REEg, VdNQ and xZvy), as well as missing discussions on wall-clock overhead (Reviewers vEd4, xZvy). In the rebuttal, the authors adequately addressed these issues by providing the necessary VCT formalization and proofs, presenting additional evaluations, and clarifying the overhead. Given the empirical results and the satisfactory resolution of the reviewers' concerns, the paper is recommended for acceptance.